# A Simple and Affordable Method to Create Nonsense Mutation Clones of p53 for Studying the Premature Termination Codon Readthrough Activity of PTC124

**DOI:** 10.3390/biomedicines11051310

**Published:** 2023-04-28

**Authors:** Chia-Chi Chen, Ruo-Yu Liao, Fang-Yu Yeh, Yu-Rou Lin, Tze-You Wu, Alexa Escobar Pastor, Danny Danilo Zul, Yun-Chien Hsu, Kuan-Yo Wu, Ke-Fang Liu, Reiji Kannagi, Jang-Yi Chen, Bi-He Cai

**Affiliations:** 1School of Medicine, I-Shou University, Kaohsiung City 82445, Taiwan; 2Department of Physical Therapy, I-Shou University, Kaohsiung City 82445, Taiwan; 3School of Chinese Medicine for Post Baccalaureate, I-Shou University, Kaohsiung City 82445, Taiwan; 4Department of Pathology, E-Da Hospital, Kaohsiung City 82445, Taiwan; 5Department of Medical Laboratory Science, I-Shou University, Kaohsiung City 82445, Taiwan; 6Department of Biomedical Engineering, I-Shou University, Kaohsiung City 82445, Taiwan; 7School of Medicine for International Students, I-Shou University, Kaohsiung City 82445, Taiwan; 8Department of Biological Science and Technology, I-Shou University, Kaohsiung City 82445, Taiwan; 9Institute of Biomedical Sciences, Academia Sinica, Taipei City 11529, Taiwan; 10Institute of Biology and Anatomy, National Defense Medical Center, Taipei City 11529, Taiwan

**Keywords:** p53, PTC124, nonsense mutation, PTC readthrough, site-directed mutagenesis

## Abstract

(1) Background: A premature termination codon (PTC) can be induced by a type of point mutation known as a nonsense mutation, which occurs within the coding region. Approximately 3.8% of human cancer patients have nonsense mutations of p53. However, the non-aminoglycoside drug PTC124 has shown potential to promote PTC readthrough and rescue full-length proteins. The COSMIC database contains 201 types of p53 nonsense mutations in cancers. We built a simple and affordable method to create different nonsense mutation clones of p53 for the study of the PTC readthrough activity of PTC124. (2) Methods: A modified inverse PCR-based site-directed mutagenesis method was used to clone the four nonsense mutations of p53, including W91X, S94X, R306X, and R342X. Each clone was transfected into p53 null H1299 cells and then treated with 50 μM of PTC124. (3) Results: PTC124 induced p53 re-expression in H1299-R306X and H1299-R342X clones but not in H1299-W91X and H1299-S94X clones. (4) Conclusions: Our data showed that PTC124 more effectively rescued the C-terminal of p53 nonsense mutations than the N-terminal of p53 nonsense mutations. We introduced a fast and low-cost site-directed mutagenesis method to clone the different nonsense mutations of p53 for drug screening.

## 1. Introduction

A nonsense mutation is a type of point mutation in the coding region that converts a sense codon into premature termination codon (PTC) TGA, TAG, or TAA [1]. Such mutations make genes translate shortened, truncated proteins, and generally the genes express low levels or no translation products through nonsense-mediated mRNA decay (NMD) [2]. Aminoglycoside drugs, such as G418 and gentamicin, as well as non-aminoglycoside drugs, such as PTC124 and its analogues, have been shown to promote PTC readthrough or inhibit NMD, thereby enabling the recoding of nonsense mutations and allowing for the re-expression of full-length proteins [3,4,5,6].

The use of aminoglycoside drugs is limited in the clinical treatment of nonsense mutation-mediated diseases due to their potential side effects, including high cellular toxicity and cochleotoxicity, which can lead to permanent hearing loss [7,8,9]. PTC124 is the drug that has been approved by the European Medicines Agency to treat nonsense mutation-mediated Duchenne muscular dystrophy. Approximately 13% of male patients with Duchenne muscular dystrophy carry a nonsense mutation in the DMD gene that expresses a protein called dystrophin [10]. The DMD gene contains a 79-exon transcript expressed in muscle that encodes a protein of 3685 amino acid residues with a molecular weight of 427 kDa [11].

PTC124 has also been planned to be used in a preclinical trial with 20 Shwachman–Diamond syndrome patients carrying nonsense mutations [12]. Shwachman–Diamond syndrome is an inherited bone marrow failure disorder [13]. About 90% of patients with SDS present mutations in the Shwachman–Bodian–Diamond syndrome (SBDS) gene [12]. In addition, around 55% of SDS patients carry a specific nonsense mutation, SBDS K62X [12]. PTC124 can restore the full-length SBDS protein in bone marrow stem cells isolated from Shwachman–Diamond syndrome patients with the K62X mutation [14]. The SBDS gene encodes a protein of only 250 amino acid residues with a molecular weight of 29 kDa [15].

Recently, our research has shown that PTC124 can also rescue nonsense mutations in three tumor suppressor genes, p53, NOTCH1, and FAT1, leading to repression of head and neck squamous cell carcinoma (HNSCC) cell proliferation [16]. The p53 gene, encoding a protein with only 393 amino acid residues, has a cDNA of around 1.2 Kb [17]. In contrast, NOTCH1 and FAT1 encode proteins with 2555 and 4588 amino acid residues, respectively, resulting in cDNAs of around 7.5 to 13.5 Kb [18,19]. Due to their large size, these genes are difficult to clone or perform site-directed mutagenesis [20]. Therefore, we focused on p53 for this study. We collected clinical cancer samples from the COSMIC database [21] (https://cancer.sanger.ac.uk/cosmic; accessed on 6 March 2023) and determined the frequency of nonsense mutations in p53. Our analysis revealed that the rate of nonsense mutations in p53 was approximately 3.8% (Appendix A). While the overall mutation rate of p53 in cancers is around 50%, the frequency of p53 mutations in all cancer samples in the database was 33.7%. This suggested that the rate of nonsense mutations in p53 among the mutated samples was about 11% (3.8/33.7) (Appendix A).

There are 201 types of p53 nonsense mutations in cancers (Appendix A). The top 10 are R213X (760 samples), R196X (507 samples), R342X (454 samples), R306X (381 samples), Q192X (214 samples), E298X (126 samples), E294X (125 samples), Q331X (115 samples), Q317X (108 samples), and W146X (93 samples). Because different drugs have different effects on different types of nonsense mutation, how to create a point mutation for each type of p53 nonsense mutation for drug screening purposes was the primary issue considered in this study.

Traditional site-directed mutagenesis consists of two blunted ends of long primers creating a mutation site in a clone [22], which has a high price of ~100 USD/per clone due to the need for high-quality long primers, purification of 25~45 bases by high-performance liquid chromatography (HPLC) or polyacrylamide gel electrophoresis (PAGE), as well as limited super-competent cells in order to avoid DNA recombination of the two blunted ends of the primers created by PCR products with nicks [20]. The other popular site-directed mutagenesis method consists of two primers oriented in the inverse direction to create a blunt-end PCR product [23] without the need for special competent cells to knock out recombined related genes, which reduces the cost. In addition, the primers only need 24~30 bases for inverse PCR and only oligonucleotide purification cartridge (OPC) purification is required [23,24]. However, this kind of inverse PCR has another limitation, since each primer requires 5′ phosphorylation for ligation due to the lack of phosphorylation on the first base of the 5′ end during DNA synthesis [23]. Therefore, this adds another cost to inverse PCR-based site-directed mutagenesis, since ordering 5′ phosphorylated primers to make a clone costs ~50 USD. The short length of the DNA (<100 bp) means that a spin column cannot be used to purify the polynucleotide kinase (PNK) reaction products when adding 5′ phosphorylation to the primers [25]. If commercial 5′ phosphorylated primers are ordered, overnight ethanol precipitation is required for PNK products with such short lengths [26]. We attempted to modify the inverse PCR-based site-directed mutagenesis protocol to avoid using 5′ phosphorylated primers and compared three different kinds of DNA proofreading polymerases used for inverse PCR.

## 2. Experimental Design

### 2.1. Materials

OPC purification primers (Genomics, New Taipei City, Taiwan) (mutated site is underlined)p53 W91X: (−) 5′-ACAGGGGTCAGGAGGGGGCT-3′ and(+) 5′-CATCTTCTGTCCCTTCCCAGAAAACCTACCA-3′p53 S94X: (−) 5′-GGGACAGAAGATCACAGGGGCCAGG-3′ and(+) 5′-TTCCCAGAAAACCTACCAGGGCAGCTAC-3′p53 R306X: (−) 5′-GGGCAGTGCTCACTTAGTGCTCCCT-3′ and(+) 5′-AACAACACCAGCTCCTCTCCCCAGC-3′p53 R342X: (−) 5′-GAAGCGCTCACGCCCACGGAT-3′ and(+) 5′-GAGATGTTCTGAGAGCTGAATGAGGCCTTGGAA-3′CloneAmp HiFi PCR Premix (TaKaRa Bio, Shiga, Japan; #639298)Phusion Plus PCR Master Mix (Thermo Scientific, Waltham, MA, USA; #F631S)Phusion High-Fidelity PCR Master Mix (Thermo Scientific, Waltham, MA, USA; #F-531S)T4 polynucleotide kinase (T4 PNK) (New England Biolabs, Hitchin, UK; #M0201S)YB Rapid Ligation Kit (Yeastern Biotech, Taipei City, Taiwan; # FYC003-100R); each kit contains yT4 DNA Ligase 100 μL (3 U/μL), 10× Ligation Buffer A, and 10× Ligation Buffer B.ECOS 9-5 Competent Cells [strain JM109] (Yeastern Biotech, Taipei City, Taiwan; #FYE707-10VL)GeneJET Plasmid Midiprep Kit (Thermo Scientific, Waltham, MA, USA; #K0481)Opt-MEM medium (Thermo Scientific, Waltham, MA, USA; #31985062)TransIT-X2 Transfection Reagent (Mirus Bio, Madison, WI, USA; #MIR6000)PTC124 (MedChemExpress, Monmouth Junction, NJ, USA; #HY-14832)Primary antibody p53 (clone DO-1; Santa Cruz, Dallas, TX, USA; #sc-126) recognized N-terminal epitope mapping amino acid residues 11–25 of human p53.Primary antibody p53 (clone PAb 122; Thermo Scientific, Waltham, MA, USA; #MA5-12453), recognizes C-terminal epitope mapping amino acid residues 370–378 of human p53.Secondary antibody PE-conjugated mouse IgG (Thermo Scientific, Waltham, MA, USA; #M30004-1)Hoechst 33342 (Tocris Bioscience, Ellisville, MS, USA; #5117) was prepared as stock solution (1 mg/mL in water)

### 2.2. Equipment

SimpliAmp Thermal Cycler (Applied Biosystems, Waltham, MA, USA; #A24811)NanoDrop ND-1000 (Thermo Scientific, Waltham, MA, USA; #ND1000WOC)ECLIPSE Ts2 inverted fluorescence microscope (Nikon, Tokyo, Japan; #094604D)

## 3. Procedure

The modified inverse PCR-based site-directed mutagenesis is shown in Figure 1.

### 3.1. Part 1: Creation of p53 Nonsense Mutation Clones (Modified Inverse PCR-Based Site-Directed Mutagenesis)

#### 3.1.1. Inverse PCR

One reaction included the following:CloneAmp HiFi PCR Premix or Phusion High-Fidelity PCR Master Mix (Thermo Scientific, Waltham, MA, USA; #F-531S) or Phusion High-Fidelity PCR Master Mix: 10 µLUpstream primer (−) 5 µM: 1 µLDownstream primer (+) 5 µM: 1 µLp53 wild-type plasmid (pcDNA 3.0 p53) 1 ng/µL: 1 µLSterilized distilled water: 7 µL

Total volume per PCR reaction: 20 µL

The PCR steps were as follows:Initial denaturation: 98 °C for 30 sDenaturation: 98 °C for 10 sAnnealing and extension: 72 °C for 2 minFinal extension: 72 °C for 5 min

Steps 2 and 3 were repeated for 35 cycles.

#### 3.1.2. T4 PNK Reaction

A 10 µL aliquot of the PCR reaction products was checked by DNA electrophoresis, appearing as a single clear DNA band of ~8 K base pairs on the gel. The remaining 10 µL of PCR reaction products was mixed with 4 µL of 10× T4 PNK buffer, 1 µL of T4 PNK, and 25 µL of sterilized distilled water, and the mixture was heated on a 37 °C hotplate for 30 min to obtain 40 µL of T4 PNK reaction products.

#### 3.1.3. Ligation

A 7 µL sample of the T4 PNK reaction products was mixed with 1 µL of 10× Ligation Buffer A, 10× Ligation Buffer B, and 1 µL of yT4 DNA Ligase (3 U/μL), and the mixture was held at room temperature for 5 min to obtain 10 µL of DNA ligation products.

#### 3.1.4. Transformation

A 3 µL sample of the DNA ligation products was mixed with 33 µL of JM109 competent cells (ECOS 9-5 Competent Cells [strain JM109] from Yeastern Biotech, Taipei City, Taiwan; #FYE707-10VL) and vortexed for 1 s before being placed on ice for 2 min. According to the manufacturer’s protocol, heat shock of ECOS 9-5 Competent Cells was only suggested for the transformation of plasmids < 6 Kb. Since our p53 clones were >6 Kb (vector 5.4 Kb + insert 1.2 Kb), we did not perform heat shock during the transformation process. We added all of the *Escherichia coli* (*E. coli*) solution to the LB agar plate with antibiotics and spread it using sterilized glass beads.

#### 3.1.5. Sequencing

One day after the LB agar plate was incubated in a 37 °C incubator, single colonies were individually selected using a sterilized tip, mixed with 5 mL of *E. coli* broth with antibiotics, and cultured for one day in a 37 °C incubator. A 3 mL sample of cultured *E. coli* solution was sent to Genomics (New Taipei City, Taiwan) to perform DNA sequencing.

### 3.2. Part 2: Transfection of p53 Nonsense Mutation Clones into p53 Null Cells

#### 3.2.1. Plasmid Isolation

One day before amplifying the *E. coli* stock with different p53 nonsense mutation clones, each plasmid clone was isolated from 5 mL of *E. coli* broth using the GeneJET Plasmid Midiprep Kit (Thermo Scientific, Waltham, MA, USA; #K0481). The NanoDrop ND-1000 (Thermo Scientific, Waltham, MA, USA; #ND1000WOC) was used to check the concentration and quality of the final plasmids.

#### 3.2.2. Liposome-Mediated Transfection

H1299 cells (1 × 10^5^) were plated into a 24-well dish one day before plasmid transfection. The following day, 0.5 µg of each p53 nonsense mutation clone plasmid was mixed with 50 µL of Opt-MEM medium (Thermo Scientific, Waltham, MA, USA; #31985062) in a 1.5 mL tube. Next, 1 µL of TransIT-X2 Transfection Reagent (Mirus Bio, Madison, WI, USA; #MIR6000) was added to the 1.5 mL tube, and the mixture was incubated at room temperature for 20 min. Each mixture was then added to a well of each dish.

### 3.3. Part 3: Adding PTC124 and Determining p53 Expression

After transfection for eight hours, the medium was replaced with fresh medium containing either DMSO only or 50 µM of PTC124 (dissolved in DMSO using 50 mM PTC124 stock solution). Following 48 h of treatment with PTC124, the medium was removed and the cells were washed twice with 1× PBS. A fixative solution (200 µL; 4% paraformaldehyde in 1× PBS) was added to each well of a 24-well dish, and the cells were incubated for 20 min. The fixative solution was then removed, and each well was washed twice with 200 µL of wash buffer (1% BSA in 1× PBS). Blocking solution (200 µL; 1% BSA, 0.2% Triton X-100 in 1× PBS) was added, and the cells were incubated for 30 min. After the blocking solution was removed, each well was washed three times with 200 µL of wash buffer. Primary antibody p53 (clone DO-1; Santa Cruz, Dallas, TX, USA; #sc-126 or clone PAb 122, Thermo Scientific, Waltham, MA, USA; MA5-12453) was diluted 1:100 in 200 µL of blocking buffer and added to each well to label the cells for two hours. Each well was washed three times with 200 µL of wash buffer. Secondary antibody PE-conjugated mouse IgG (Thermo Scientific, Waltham, MA, USA; # M30004-1) and Hoechst 33342 (Tocris Bioscience’s, Ellisville, MS, USA; #5117) stock solution (1 mg/mL) were added to 200 µL of blocking buffer to stain the cells for 30 min in the dark. Finally, each well was washed three times with 200 µL of wash buffer, and cell images were captured using an ECLIPSE Ts2 inverted fluorescence microscope (Nikon, Tokyo, Japan; #094604D).

## 4. Results

PTC124 has a different readthrough efficiency in different types of pre-stop codons. The efficiency of PTC124 for pre-stop codon readthrough is TGA > TAG > TAA [27]. Nonsense mutations of p53 clones in the COSMIC database with the same type of pre-stop codon were searched. All four of the p53 mutations selected in this study contained a TGA pre-stop codon and were clinically relevant in lung cancer cases with frequencies ranging from 2% to 23% according to the COSMIC database (Table 1). We generated these nonsense mutations and corresponding clones using a modified inverse PCR-based site-directed mutagenesis method (see Figure 1). The inverse PCR products and sequencing results of the p53 nonsense mutation clones are presented in Figure 2 and Figure 3, respectively.

Previously, 50 µM of PTC124 could strongly re-express p53 in SAS HNSCC cells with the E336X mutation, which contains the TAG pre-stop codon [16]. The flowchart of this research and the map of p53 nonsense mutation clones are shown in Figure 4. It was found that 50 µM of PTC124 could only induce p53 expression in H1299-R306X and H1299-R342X clones but not in H1299-W91X and H1299-S94X clones (Figure 5 and Figure 6). Our data demonstrated that PTC124 was more likely to be effective in rescuing the C-terminal of p53 nonsense mutations than the N-terminal of p53 nonsense mutations (Figure 4, Figure 5 and Figure 6).

## 5. Discussion and Conclusions

In this study, our findings showed that PTC124 was more effective in promoting the readthrough of the C-terminal of nonsense mutations of p53 than the N-terminal of nonsense mutations of p53 (as shown in Figure 4, Figure 5 and Figure 6). This observation was consistent with similar results found with the HERG gene, in which gentamicin was responsive to R1014X (TGA-type pre-stop codon) and W927X (TGA-type pre-stop codon) but not to R863X (TGA-type pre-stop codon) and E698X (TAG-type pre-stop codon) [28]. Furthermore, PTC124 was able to re-express HERG with W927X, whereas R863X and E698X remained unresponsive [29].

In addition to the location of the nonsense mutation in the coding region, the efficacy of readthrough can vary depending on the type of drug used and the type of pre-stop codon. Studies have shown that PTC readthrough efficiency of PTC124, G418, and gentamicin follows the order of TGA > TAG > TAA [27,30], whereas clitocine has a different order of TAA > TGA > TAG [31]. Furthermore, 2,6-diaminopurine (DAP) has been shown to be effective only for TGA readthrough and has no effect on TAA or TAG [32]. It is not feasible to obtain all of the relevant cell lines with the same nonsense mutation sites of specific genes in clinical samples for drug screening. For instance, while there are 201 types of p53 nonsense mutations in cancers (Appendix A), only 44 types of p53 nonsense mutations in cell lines collected in the COSMIC Cell Lines Project are available (Appendix A) (https://cancer.sanger.ac.uk/cell_lines; accessed on 6 March 2023) [21]. Therefore, a fast and cost-effective approach to clone each clinically relevant nonsense mutation of the p53 gene and express them in p53 null expression cells provides a simple and efficient drug screening method. This protocol can also be useful for other tumor suppressor genes with high nonsense mutation rates.

Compared to traditional methods of site-directed mutagenesis, which require high-quality blunt-end long primers or expensive commercial 5′ phosphorylated primers for inverse PCR based site-directed mutagenesis, this study introduces a modified inverse PCR-based site-directed mutagenesis protocol that is both fast and cost-effective. This method utilizes short OPC-grade primers, making it easy to clone different types of nonsense mutations for drug screening. This approach was successfully employed to screen for readthrough activity in a variety of genetic diseases or cancers involving critical gene(s) harboring specific sites of nonsense mutations.

## Figures and Tables

**Figure 1 biomedicines-11-01310-f001:**
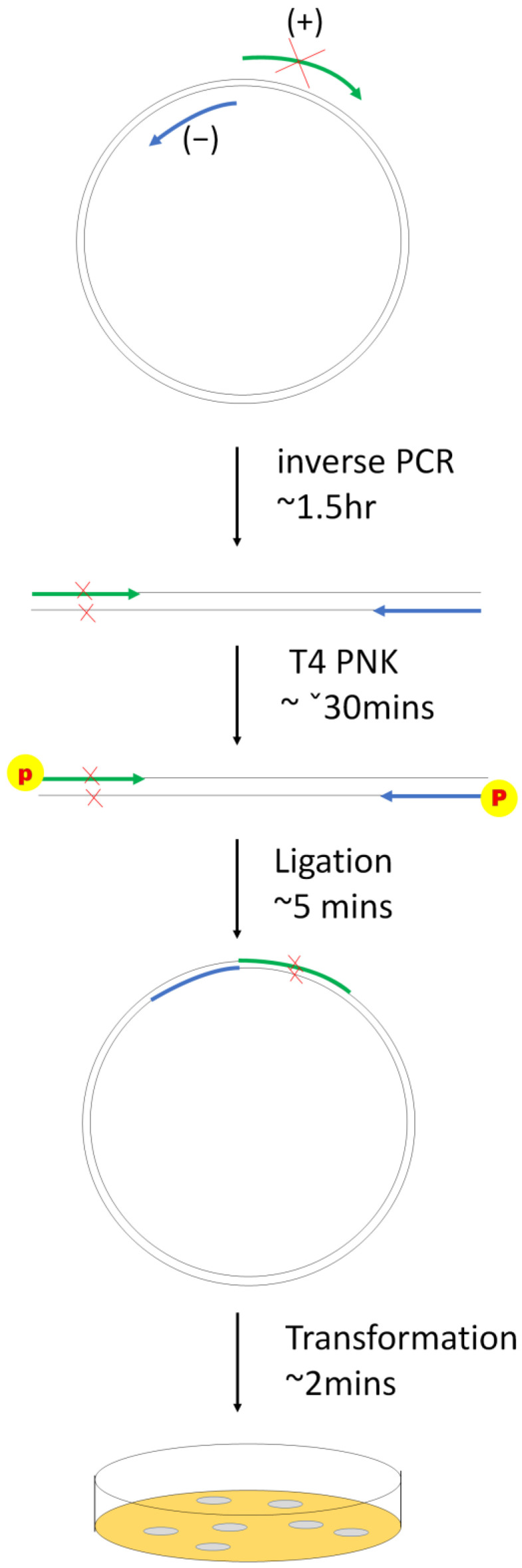
Modified inverse PCR-based site-directed mutagenesis. The detailed steps are mentioned in protocol 3.1. The mutated site can be located in the middle (+) or (−) stream of the primer. The ligation and transformation steps should not be longer than 10 min because this dramatically reduced the number of colonies on the LB agar plate for sequencing.

**Figure 2 biomedicines-11-01310-f002:**
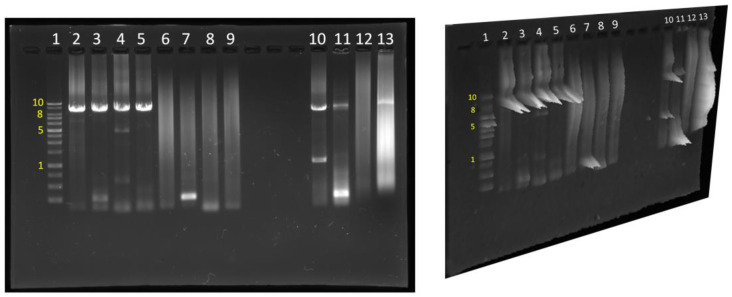
The inverse PCR products of p53 nonsense mutation clones. Each number represents each band. Band 1: Kb marker (10: 10 Kb; 8: 8 Kb; 5: 5 Kb; 1: 1 Kb). Bands 2–5: PCR products of W91X, S94X, R306X, and R342X of p53 clones were amplified using CloneAmp HiFi PCR Premix. Bands 6–9: PCR products of W91X, S94X, R306X, and R342X of p53 clones were amplified using Phusion Plus PCR Master Mix. Bands 10–13: PCR products of W91X, S94X, R306X, and R342X of p53 clones were amplified using Phusion High-Fidelity PCR Master Mix.

**Figure 3 biomedicines-11-01310-f003:**
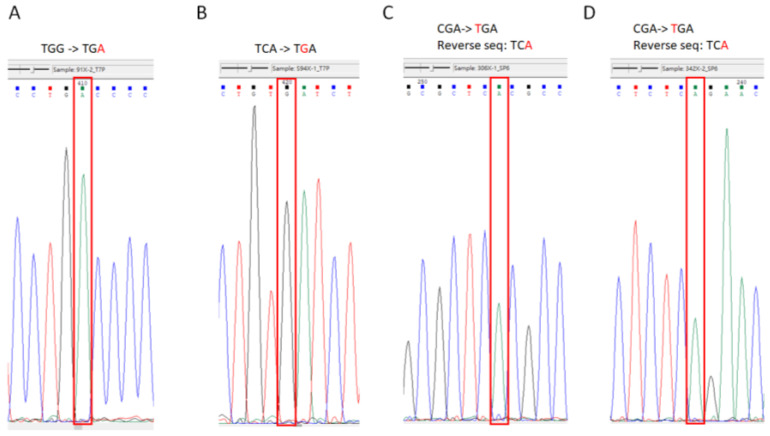
The sequencing of p53 nonsense mutation clones. The sequencing results of a selected single colony from bands 2–5 of Figure 2. (**A**) W91X (TGG to TGA, the area enclosed in the red rectangle is the changed A base), (**B**) S94X (TCA to TGA, the area enclosed in the red rectangle is the changed G base), (**C**) R306X (CGA to TGA [reversed sequence is TCA], the area enclosed in the red rectangle is the changed A base), and (**D**) R342X (CGA to TGA [reversed sequence is TCA], the area enclosed in the red rectangle is the changed A base).

**Figure 4 biomedicines-11-01310-f004:**
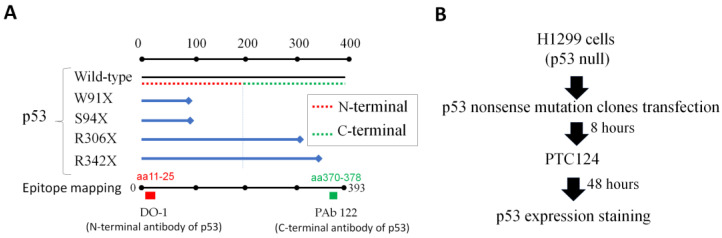
Flowchart of this research. (**A**) The four nonsense mutation clones studied in this research. The p53 N-terminal and C-terminal amino acids are represented by red and green dotted lines, respectively. Blue lines and diamonds indicate the p53 mutants and their pre-stop site. The p53 N-terminal antibody (DO-1) could recognize amino acid residues 11–25, whereas the p53 C-terminal antibody (PAb 122) could recognize amino acid residues 370–378. (**B**) Four of the p53 nonsense mutation clones were transfected into H1299 cells. A 50 µM volume of PTC124 was added to the cells after transfection for 8 h of p53 nonsense mutations. The p53 expression was checked after incubation for 48 h.

**Figure 5 biomedicines-11-01310-f005:**
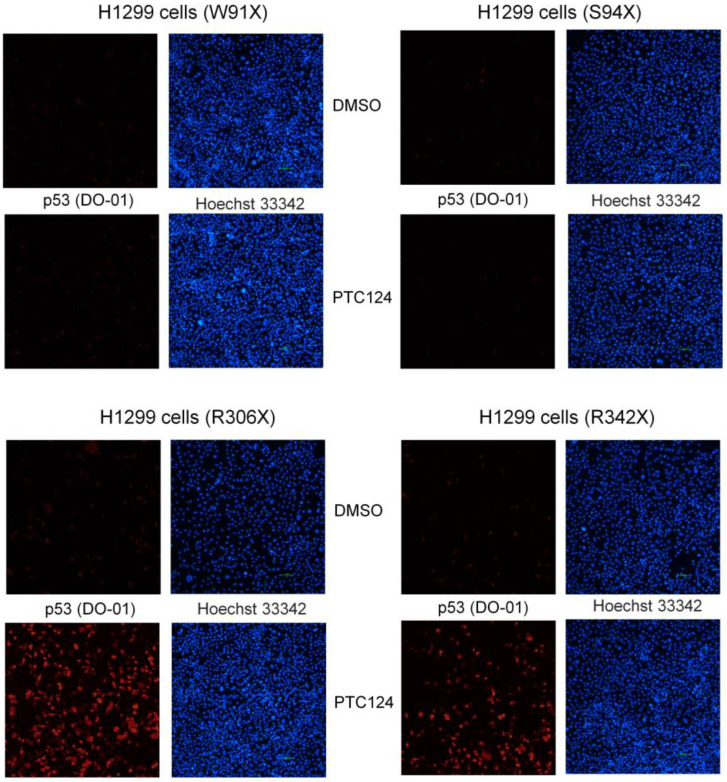
The expression level of p53 was assessed using the p53 N-terminal antibody (DO-1) following treatment with either DMSO or PTC124 in four different p53 nonsense mutation clones. It was observed that PTC124 (50 μM) was able to induce re-expression of p53 as red color in H1299-R306X and H1299-R342X clones but not in H1299-W91X and H1299-S94X clones. Hoechst 33342 was used to label the nucleus as blue color.

**Figure 6 biomedicines-11-01310-f006:**
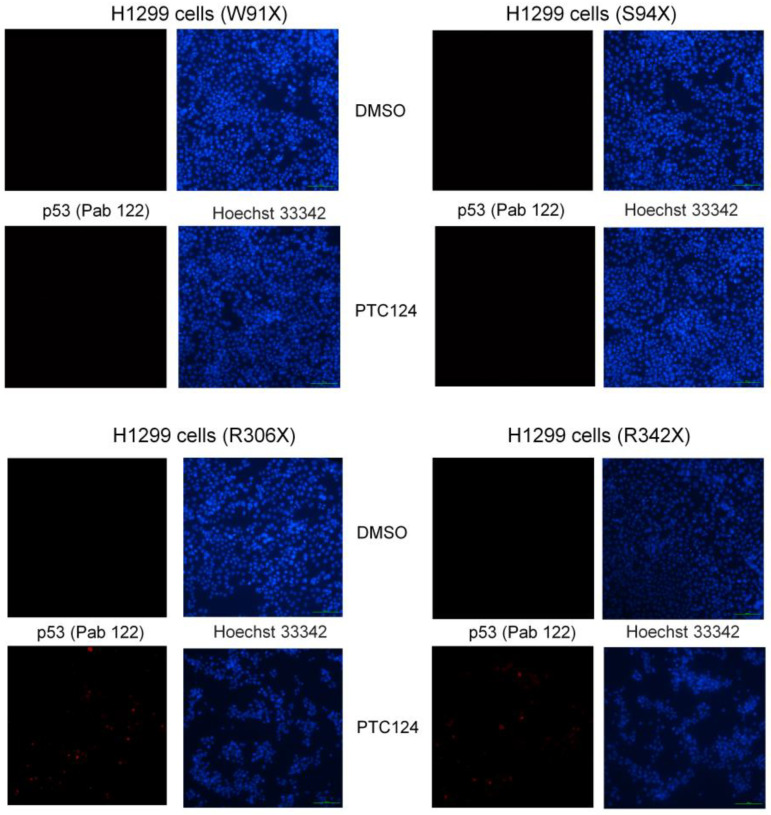
The expression level of p53 was assessed using the p53 C-terminal antibody (PAb 122) in the presence of either DMSO or PTC124 treatment across four different p53 nonsense mutation clones. Our results showed that PTC124 (50 μM) induced re-expression of p53 as red color in H1299-R306X and H1299-R342X clones but not in H1299-W91X and H1299-S94X clones. Nuclear staining was performed using Hoechst 33342 as the blue color.

**Table 1 biomedicines-11-01310-t001:** The list of nonsense mutations of p53 studied in this research. All mutation types were selected from the COSMIC database [21] and the relative number of lung cancer samples are listed. Four different p53 nonsense mutations had TGA-type pre-stop codons.

Nonsense Mutation Gene	Protein Mutation Site	DNA Mutation Site	Original Codon	Mutated Pre-Stop Codon	Lung Cancer Samples/All Type Samples	COSMIC ID
*P53*	p.W91X	c.273G > A	TGG	TGA	13/69 (23%)	COSM44492
*P53*	p.S94X	c.281C > G	TCA	TGA	1/11 (9%)	COSM45653
*P53*	p.R306X	c.916C > T	CGA	TGA	6/376 (2%)	COSM10663
*P53*	p.R342X	c.1024C > T	CGA	TGA	22/453 (5%)	COSM11073

## Data Availability

Data are available upon reasonable request to the submitting author. Data were obtained from the COSMIC database, which is freely available for non-commercial users.

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
