# Peer review of "A Simple and Affordable Method to Create Nonsense Mutation Clones of p53 for Studying the Premature Termination Codon Readthrough Activity of PTC124"

_biomedicines, 2023, doi:10.3390/biomedicines11051310_

Round 1

Reviewer 1 Report

The reviewed study is a reasonable attempt to retrieve suppressor activity of TP53 in cancer (not specified). TP53 gene is highly mutable in many cancers generated by chemical agents present  in human surrounding (occupational and environmental exposure) separately or in mixtures  including tobacco smoke. Hence, the aim of the study would be beneficial for cancer patients.

The authors studied PTC124 to restore activity of TP53. Title seems to suggest a huge amount of data concerning a variety of TP nonsense mutations. Instead the authors have chosen 4 mutation location close to N- or C-terminus of the gene. I am not blaming the authors for such narrowing of the target as the findings discovered a difference of both termini in rescuing  gene expression.  The finding is an interesting item and workable for further (clinical? ) studies).

A background of the study is well presented. The experiments are precisely described and documented.

Concerning high toxicity of aminoglycoside drugs I would mention  ototoxity responsible for harmfull side effects of chemotherapy.

Minor remarks:  All but two literature citations have full (traditional and electronic) data. Article pages are missing in citation 6 and 15.

Very minor correction needed.

Reviewer 2 Report

The manuscript describes a simple and affordable method to generate point mutations in the coding sequences of genes cloned in bacterial vectors. The matter of cost is important in the context of the research goals of the authors, who plan to screen, which nonsense mutations are amenable to the induction, by the compound named PTC124, of stop codon read-through. The authors tested their method by generating premature stop codons in the coding sequence of p53 tumor suppressor gene. These naturally occurring but artificially generated stop codons were tested in p53-null cells exposed to PTC124. This compound induced the read-through in stop codons located near the C-terminal part of the coding sequence but had no observable influence on the stop codons located near the N-terminus of the p53.

I have the following critical comments concerning this manuscript. I leave to the editors the judgement how important they are.

1.       It is hard to understand what is the major innovation introduced by the authors in the method of mutagenesis. It is hinted at in the last sentence of the Results (lines 90-92). As far as I can understand, the novelty is that the authors did not use the expensive and hard to handle primers with phosphates at 5’-ends but instead introduced this phosphate by a reaction with T4 kinase, which phosphorylated the PCR products.

2.       The above problem may result from low quality of data presentation. There are parts of the manuscript, which I do not understand, e.g., lines 63-65, lines 208-210.

3.       Some acronyms are not explained, e.g., HNSCC (line 58) ). I can guess what it means but it should be explained.

4.       There are some repetitions, e.g., line 60: “FAT1 has 4588 aa amino acids”, similar problem lines 59-60.

5.       There are some direct tandem insertions in the text: see lines 250-251.

6.       As far as I can judge the English language does not require major correction but the style of presentation and clarity of text definitely does. By the way, in the Authors contributions section there are 5 authors who reviewed and edited the manuscript – the next time they have to do a better job.

7.       In the section 3.1.1. Transformation (lines 166-168) – the authors described the method of introduction of plasmid into bacterial cells. They mention the incubation on ice for 2 min. (standard procedure), but there was no heat shock (usually 37oC for less than 1 min.) – is it an omission or this method does not require the heat shock? Because the description of the modified technique as the major goal of this paper, the authors should make sure that the description of the technique is flawless, e.g., the omission of the heat shock in the transformation can lead to total failure of the experiment.

8.       The conclusion “Our data showed that the C-terminal of nonsense mutations of p53 were more effectively retrieved by PTC124 than the N-terminal of nonsense mutations (Figures 4 and 5).” is a little too strong considering that the authors tested only 4 mutations. In this particular case the sentence is correct but extending the conclusion to other N or C terminal mutations may be risky.

Reviewer 3 Report

This is a well-described, and truly simple and affordable technique that might be useful for the researchers working in the p53 field.

Author Response

We are sincerely grateful.

Reviewer 4 Report

This study describes a methodology that more efficiently introduces mutations into the p53 gene for accessing readthrough strategies that suppress translation termination at premature termination codons. This strategy was tested with four different p53 nonsense mutations, where the p53 constructs were transfected into p53 null cells that were treated with the readthrough compound PTC124 and  examined for p53 expression as a measure of readthrough efficiency. While the methodology to generate mutations in p53 appears to be sound and produced clones of the appropriate construct, the data regarding readthrough of p53 by PTC124 is cannot be interpreted due to errors in the experimental design.  

The following must be addressed:

Major changes:  According to the Santa Cruz information on the #sc-126 antibody, the antibody that was used to detect p53 protein via immunofluorescence recognizes an epitope in the N-terminus of p53 (amino acids 11-25). This means that this antibody recognizes both truncated p53 (no readthrough) and full-length p53 (+ readthrough of nonsense mutation). Therefore, the immunofluorescence data cannot distinguish between truncated and full-length p53 protein, and therefore, cannot serve as a specific measure for readthrough. The authors need to either obtain an antibody to the C-terminus of p53 that will only recognize p53 resulting from readthrough or even better, perform western blotting so that the molecular weight of p53 can be observed to distinguish between truncated p53 and full-length p53.

Minor changes:

1) Page 2, line 58, please indicate what the abbreviation HNSCC stands for.

2) More details should be provided for the experimental design...i.e., the epitope that is recognized by p53 antibody, what vehicle was used to administer PTC124.

3) A dose response curve for PTC124 should be performed to make sure the optimal dose was used for this cell line.

4) The last paragraph of the discussion that describes the pros and cons of previous mutagenesis strategies needs to be edited for clarity. It seems that some run-on sentences are present and some grammatical changes are needed.

Round 2

Reviewer 4 Report

It appears that my previous concerns have been addressed.